# Brain-tuning Improves Generalizability and Efficiency of Brain Alignment in Speech Models

**Omer Moussa**
Max Planck Institute for Software Systems
Saarbrücken, Germany
omoussa@mpi-sws.org

**Mariya Toneva**
Max Planck Institute for Software Systems
Saarbrücken, Germany
mtoneva@mpi-sws.org

## Abstract

Pretrained language models are remarkably effective in aligning with human brain responses elicited by natural language stimuli, positioning them as promising model organisms for studying language processing in the brain. However, existing approaches for both estimating and improving this brain alignment are participant-dependent and highly affected by the amount of data available per participant, hindering both generalization to new participants and population-level analyses. In this work, we address these limitations by introducing a scalable, generalizable brain-tuning method, in which we fine-tune pretrained speech language models to jointly predict fMRI responses from multiple participants. We demonstrate that the resulting brain-tuned models exhibit strong individual brain alignment while generalizing across participants. Specifically, our method leads to 1) a 5-fold decrease in the amount of fMRI data needed to predict brain data from new participants, 2) up to a 50% increase in the overall brain alignment, and 3) strong generalization to new unseen datasets. Furthermore, this multi-participant brain-tuning additionally improves downstream performance on semantic tasks, suggesting that training using brain data from multiple participants leads to more generalizable semantic representations. Taken together, these findings demonstrate a bidirectional benefit between neuroscience and AI, helping bridge the gap between the two fields. We make our code and models publicly available at https://github.com/bridge-ai-neuro/multi-brain-tuning.

## 1 Introduction

Language models (LMs) substantially align with human brain responses elicited during natural language processing, such as functional magnetic resonance imaging (fMRI) signals recorded while participants listen to or read stories [Toneva and Wehbe, 2019, Schrimpf et al., 2021, Goldstein et al., 2022, Millet et al., 2022, Oota et al., 2024, Moussa et al., 2025]. This intriguing correspondence has positioned pretrained LMs as promising model organisms for studying human cognition, particularly language comprehension [Toneva, 2021]. Such models can offer new insights into how the brain represents and processes complex linguistic information.

However, despite the growing success of pretrained models in predicting brain activity, current brain alignment approaches face significant limitations. Existing methods are data inefficient, requiring extensive fMRI data from a new participant to estimate alignment with a specific LM [Antonello et al., 2024]. Additionally, these methods are participant-dependent, which limits generalization to new participants. Even methods that incorporate brain data directly into the model during training remain participant-dependent [Moussa et al., 2025, Vattikonda et al., 2025]. As a result, leveraging these models to study language processing across populations remains a challenge. Addressing these issues is critical to advancing the use of language models as proxies for human language processing.

39th Conference on Neural Information Processing Systems (NeurIPS 2025).

To overcome these challenges, we propose a novel approach, designed to improve the efficiency and generalization of brain alignment in speech models. Our method fine-tunes pretrained speech language models to jointly predict brain responses from multiple participants who are exposed to the same naturalistic speech stimuli. By pooling data across individuals, this brain-tuning approach leverages common patterns in language processing, reducing the dependency on large amounts of participant-specific data. This scalable strategy enables the creation of brain-tuned models that not only align well with individual brain responses but also generalize effectively to new participants.

Our extensive experiments demonstrate the multiple benefits of brain-tuning. First, our method reduces the amount of fMRI data required to achieve reliable brain alignment for new participants by a factor of five, significantly lowering the required fMRI data for a robust estimate of brain alignment. Second, brain-tuning yields up to a $50\%$ improvement in overall brain alignment, indicating that jointly training on data from multiple participants enhances model robustness and prediction quality. Third, our approach improves brain alignment on novel stimuli and participants, indicating a strong generalization ability. Notably, brain-tuned models also improve downstream performance on semantics-related tasks relative to their pretrained counterparts, opening up the possibility of practical integration of brain-tuned speech models into speech processing pipelines without compromising linguistic utility. We hope that our extensive experiments on multiple pretrained models, as well as comprehensive ablation studies, will help establish best practices for brain-tuning.

The proposed brain-tuning method provides a stepping stone towards scalable, participant-agnostic brain alignment, facilitating more inclusive and generalizable models of human language processing. By enabling the use of pretrained language models in population-level neuroscience studies, our work bridges the gap between advanced computational techniques and the study of human cognition, providing a robust foundation for future interdisciplinary research. We make all code and trained models publicly available to facilitate reproducibility and further research in brain-tuning.

## 2 Related Work

A growing number of studies investigate the degree of alignment between language-evoked brain activity and text-based language models [Wehbe et al., 2014, Jain and Huth, 2018, Toneva and Wehbe, 2019, Abdou et al., 2021, Schrimpf et al., 2021, Toneva et al., 2022a,b, Antonello et al., 2021, Oota et al., 2022, Merlin and Toneva, 2022, Antonello et al., 2024] and speech-based language models [Millet et al., 2022, Vaidya et al., 2022, Tuckute et al., 2023, Oota et al., 2023, 2024, Chen et al., 2024]. These works rely on participant-specific brain encoding models, as brain representations exhibit substantial inter-subject variability due to anatomical and functional differences. Moreover, accurately estimating brain alignment has been shown to require a large amount of per-participant data [Antonello et al., 2024]. This participant-specific nature severely limits model generalization across individuals, particularly with modalities like fMRI, EEG, and MEG.

Previous work has proposed methods to unify multiple participants into the same space [Chen et al., 2015, Haxby et al., 2020, Nastase et al., 2020, Beliy et al., 2025]. However, these approaches need huge datasets and are typically not focused on the language and speech domains. Moreover, they do not leverage the existing powerful pretrained models and instead train brain encoding models from scratch. Additional work that focuses on brain *decoding* (i.e., predicting what stimulus a participant observes from their brain response) has also made strides towards unifying brain responses from multiple participants. This is typically achieved by training participant-specific projection networks [Tang and Huth, 2025, Jayalath et al., 2024, Défossez et al., 2023]. Such methods usually focus on learning low-dimensional features, scale poorly with many participants, and require participants to share the stimuli. In contrast, our method leverages pretrained language models for improved efficiency and performance for encoding models, with a focus on language and speech in the brain.

A promising recent direction for improving brain encoding involves fine-tuning pretrained language models using brain fMRI responses for naturalistic stimuli (i.e., brain-tuning [Moussa et al., 2025] and BrainWavLM [Vattikonda et al., 2025]). Brain-tuning demonstrated that fine-tuning with brain data enhances a speech model's brain alignment to semantic cortical regions and also improves the model's semantic performance on downstream tasks [Moussa et al., 2025]. Similarly, BrainWavLM uses low-rank adaptation (LoRA) to fine-tune a pretrained speech transformer on naturalistic fMRI data, showing improvement in brain alignment for a training participant and a held-out participant [Vattikonda et al., 2025]. While these methods illustrate that limited neural data can significantly

enhance the alignment of pretrained model representations, they are built per participant and do not learn from brain data from multiple sources, limiting their scalability and generalization. Our method overcomes these limitations, enabling the model to be jointly trained using responses from multiple participants. By jointly brain-tuning models across multiple participants, we increase efficiency and enhance generalization, reducing the need for huge per-participant data and facilitating robust and scalable models for brain alignment and cognitive neuroscience research. Additionally, we explicitly investigate the impact of our brain-tuning on the amount of fMRI data needed for brain encoding models, as well as the impact of the tuning data size on generalization to novel participants and data.

# 3 Methods

## 3.1 Pretrained Speech Models

To evaluate different starting points for brain-tuning, we used two popular pretrained transformer-based speech model families: Wav2Vec2.0 [Baevski et al., 2020] and HuBERT [Hsu et al., 2021]. We selected comparable versions of the models, each with 90M parameters, 12 transformer layers, an embedding dimension of 768, and a 20ms input token length. Both models are self-supervised and were pretrained to predict masked segments on a 960-hour audio dataset that is independent of the fMRI datasets that we use for developing and testing our brain-tuned models.

## 3.2 Naturalistic Brain Datasets

### 3.2.1 Datasets Details

For brain-tuning and evaluation, we use the **Moth Radio Hour** dataset [LeBel et al., 2024], which is the largest per-participant fMRI dataset that is publicly available. This dataset consists of fMRI recordings of 8 participants who listened to autobiographical stories from the Moth Radio Hour podcast. Three participants listened to 84 stories ($\approx$16.1h of audio for each participant), while the rest listened to 27 stories ($\approx$6.4h of audio). fMRI images were acquired every 2s (TR = 2.0s).

To test cross-dataset generalization, we use a subset of the **Narratives** fMRI dataset [Nastase et al., 2021], in which 16 participants listened to a 56-minute fictional short story (with TR = 1.5s). This subset provides a suitable setting to test generalization to a new dataset, as it has many participants and less per-participant data than the Moth Radio Hour dataset.

### 3.2.2 Spatial Alignment of fMRI Data Across Participants

A key challenge in multi-participant brain-tuning is the anatomical differences across individuals, leading to variations in brain size (and hence the number of fMRI voxels), surface geometry, and region boundaries. This makes it challenging to train jointly or make predictions for a new participant. Moreover, using the whole cortex for fine-tuning, as done in [Moussa et al., 2025, Vattikonda et al., 2025] limits our ability to control what areas are included during fine-tuning. We solve this by spatially aligning participants.

To spatially align participants and be able to parse specific regions of interest (ROIs), we project each participant's data to a common cortical surface with FreeSurfer v7. We then use the cerebral parcellation atlas from [Glasser et al., 2016] to parse auditory regions (A1 through A4) and late language ROIs (e.g., bilateral inferior frontal gyrus, angular gyrus, anterior and posterior temporal lobes, and middle frontal gyrus). The full list of ROIs and their functions is provided in Supp.A.1.

## 3.3 Brain-tuning Approach

In this section, we elaborate on our approach and training details for fine-tuning speech models with fMRI data from multiple participants (i.e., Multi-brain-tuning) as well as the comparison baselines, namely brain-tuning with a single participant, LLM-tuning, and Stimulus-tuning (Sec.3.3.4).

### 3.3.1 Data Preparation

We follow Moussa et al. [2025], Antonello et al. [2024] to preprocess the stimulus and brain data for training. We divide the audio stimulus into snippets of 2s. We then concatenate each snippet

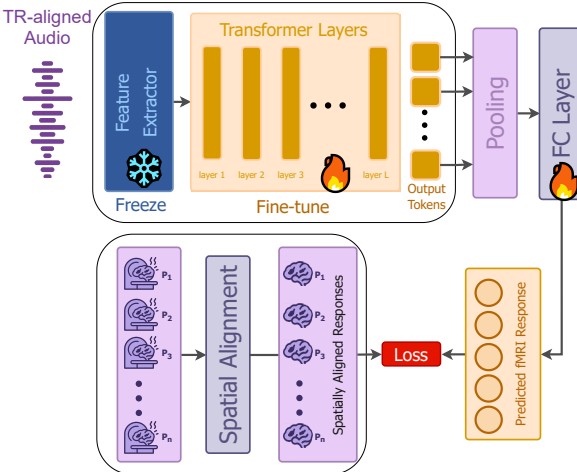

Figure 1: Multi-brain-tuning Approach. Given participant responses, we project them to a common space (see Sec.3.2.2). The aligned responses of multiple participants are then used to fine-tune the pretrained model with low-rank updates, binding by the shared stimulus as detailed in Sec.3.3.2.

with the preceding $8s$ (4TRs) to account for the fMRI hemodynamic response delay. This results in a paired audio-fMRI dataset, with each sample having a $10s$ audio clip and its one corresponding fMRI TR. fMRI responses are then spatially aligned and parcellated as described in Sec.3.2.2. Parsing the desired auditory and language ROIs results in 30K voxels across both hemispheres.

### 3.3.2 Brain-tuning with Multiple Participants

After preparing the training data, we perform brain-tuning across multiple participants (**Multi-brain-tuning** shown in Fig.1). To achieve this, we add an average pooling layer followed by a unified projection head on top of the speech model. Given a stimulus batch $S$ that is sampled contiguously from the audio, and the corresponding fMRI responses from participants $P_{1...n}$, we fine-tune the pretrained model to predict the participants' fMRI responses. This involves computing and backpropagating the training loss on each $(S, P_i)$ pair independently and sequentially.

This training strategy encourages the model to learn more generalizable representations and increases robustness to noise inherent in individual responses. We observed that it outperforms methods that predict the averaged fMRI response across participants or those that average the loss across participants, possibly because these alternatives risk discarding unique, informative signals from individual responses. Furthermore, we found using a unified projection head on top of FreeSurfer ROIs to be better than other alternatives like shared response modeling [Chen et al., 2015] and participant-specific projection heads (as in [Défossez et al., 2023]). Detailed comparisons are provided in Supp.A.3.

Note that our method does not require all participants to have listened to the same stimulus set. Instead, we use each stimulus as an anchor during training, presenting the model with all available fMRI responses for that stimulus consecutively. This design makes the method robust to datasets where participants have varying levels of stimulus overlap. Importantly, we find that the critical factor for successful Multi-brain-tuning is having sufficient tuning data, as discussed in Sec.4.2.

We test multiple training objectives (see Sec.4.4), and conduct our final set of experiments with the best-performing one–$L_2$ reconstruction objective– as we observed that it scales better than the alternatives. Lastly, for easier scalability with more data, we reduce the number of trainable parameters using a low-rank adaptation (LoRA) over the pretrained speech models [Hu et al., 2022].

Our approach removes the limitation of per-participant fine-tuning, as in previous work [Moussa et al., 2025, Vattikonda et al., 2025] and allows the brain-tuned model to integrate information across participants due to the shared projection head. The method is also adaptable to multiple datasets.

### 3.3.3 Training details

We train the Multi-brain-tuned model using fMRI responses from the three participants with the most data from the Full Moth Radio dataset, and the rest are held out for evaluation. During fine-tuning, we update the LoRA parameters and the projection head while keeping the feature extractor frozen.

We use a LoRA rank $= 8$, which corresponds to $0.625\%$ of the total model parameters. Increasing the rank beyond 8 did not help the model (Sec.4.4). We used a learning rate of $1 \times 10^{-4}$ with a $10\%$ warmup period and a linear decay. We split the fMRI stories into 2 validation stories, 1 held-out test story (exclusively used for evaluation and never during training), and the remaining 81 stories for training. At tuning time, we use a batch size of 128 samples of (audio, fMRI response) pairs (see Sec.3.2) and train the model for 30 epochs. The training is stopped when the validation loss saturates or begins to diverge. Training takes approx. 6h on two NVIDIA A40 48GB GPUs.

### 3.3.4 Comparison Baselines

**Single-brain-tuned**. The most important baseline is the brain-tuned model with data from a single participant. We do this by limiting the data to a single source and carrying out the same method and training settings (Fig.1 and Sec.3.3.3). We train $n$ Single-brain-tuned models (one for each training participant), and report their average performance when evaluating on held-out participants.

**LLM-tuned**. An alternative way to improve a speech model is to fine-tune using representations from an LLM that encode rich semantics [Moussa et al., 2025]. Specifically, we replace the brain responses with representations obtained from LLama2-7B (see Supp.B.3 for more details).

**Stimulus-tuned**. This baseline aims to measure the benefits of brain-tuning against simply fine-tuning with the stimulus audio. We use the same pretraining self-supervised objective ([Baevski et al., 2020, Hsu et al., 2021]) to fine-tune the model with the stimulus set (see Supp.B.3 for more details).

### 3.3.5 Ablations

The training objective and the number of trainable parameters can largely affect how a model can learn from noisy fMRI data. Moreover, some objectives may be effective when the training data size is small but scale poorly and vice versa. We investigate these effects by performing ablations in our brain-tuning approach. Specifically, we vary the training objective and LoRA Ranks. We test 3 objectives: the $L_2$ loss, as done in [Moussa et al., 2025], Spatial Correlation Loss adapted from [Vattikonda et al., 2025], and a combined Cosine Similarity $+ L_2$ loss. The exact formulation of the losses can be found in Supp.A.3. To test the effect of the number of trainable parameters, we compare different LoRA ranks, measuring how the ranks affect performance and scaling.

## 3.4 Evaluation

We evaluate multiple aspects of Multi-brain-tuning: efficiency, generalization, and downstream performance. Specifically, a successful brain-tuning approach should: (1) require less data to achieve reliable brain alignment for unseen participants (improved data efficiency), (2) yield higher brain alignment on new participants and datasets (improved generalization) when tuned with more data, and (3) result in no substantial degradation of the downstream utility of the model.

### 3.4.1 Estimating Brain Alignment

We compute brain alignment using standard voxel-wise encoding models. We follow Vaidya et al. [2022], Oota et al. [2024], Moussa et al. [2025] in preparing the dataset needed for evaluation (estimating the brain alignment). First, we extract speech features via a 16.0s sliding window (stride $= 0.1$s) over the audio stimulus $S$. We feed these segments into the speech model and retain the representations of the final token. Then, we interpolate these features with a Lanczos filter to match the fMRI acquisition rate. Finally, we concatenate the features for the preceding 10s to account for the hemodynamic delay. The concatenated features are used to train the voxel-wise encoding model.

We carry out this voxel-wise encoding by learning a linear function per-voxel on the concatenated features. We use ridge regression to fit the model on the training portion of the encoding dataset, and the ridge parameter is chosen via cross validation. The encoding performance is evaluated on the held-out test set via Pearson correlation.

To estimate the alignment for the language areas, we first normalize the obtained voxel-wise correlation on the test set by the estimated voxel-wise noise ceiling (the maximum explainable variance, Supp.A.2). After this normalization, we compute the mean of the voxels belonging to the language ROIs (detailed in Supp.A.1). This serves as a standardized measure for brain alignment since it is computed relative to the estimated explainable variance in the brain region.

In all experiments, normalized brain alignment is reported as an average across the upper-middle layers of the corresponding model. These layers were selected because they have been shown to best align with language regions [Antonello et al., 2024, Oota et al., 2024, Vaidya et al., 2022]. We report the mean of this alignment across participants and the standard error of the mean in all figures.

### 3.4.2 Efficiency of Brain Alignment

The brain alignment of the existing pretrained language models is poor when the amount of data used to train the voxel-wise encoding is small [Antonello et al., 2024]. Hence, these models require a large per-participant data to achieve a good brain alignment. A model that's more efficient would need much less data to obtain the same good brain alignment.

To investigate this efficiency aspect of our Multi-brain-tuned model, we quantify the amount of encoding data needed to match the pretrained performance. We do this by gradually reducing the encoding data size, then computing the brain alignment of the voxel-wise encoding model trained on the reduced data. At each fraction of data, we compare alignment to the pretrained counterpart fitted on the full encoding dataset. We report this for training participants (the ones used for brain-tuning) and for held-out participants (unseen during brain-tuning).

In addition to the Multi-brain-tuned model, we also test the efficiency of the Single-brain-tuned and the pretrained models. Due to shared information among participants, it is expected that the Single-brain-tuned model will be more efficient than its pretrained counterpart on the held-out participants. However, we still expect the Multi-brain-tuned one to outperform both because it could better leverage shared and general information learned from multiple participants.

### 3.4.3 Generalizability of Brain Alignment

A generalizable brain-tuned model should improve brain alignment with held-out participants and out-of-distribution datasets by leveraging the shared general information it learned during brain-tuning using multiple participants. To investigate this generalization aspect, we examine a) how brain alignment changes when we vary the size of the data used for brain-tuning and b) the alignment improvement on a completely different dataset.

We expect the model to generalize when it's trained on a diverse enough dataset that also has a sufficient amount of data per participant. This is due to the noise in fMRI data, which necessitates both scale per participant and diversity to learn meaningful structure. For comparison, we similarly test the Single-brain-tuned models.

To test cross-dataset generalization, we use the subset of 16 participants from the Narratives dataset detailed in Sec.3.2.1 and estimate brain alignment (as in Sec.3.4.1) on a held-out $20\%$ of this data. As a strong baseline, we also estimate the brain alignment of a **Narratives-Multi-brain-tuned** model, a Multi-brain-tuned model on the same 16 participants used for testing.

### 3.4.4 Downstream Performance

Previous work has shown that Brain-tuning could benefit downstream performance [Moussa et al., 2025], but it didn't show whether it scales when the amount of brain-tuning data is beyond a single participant. Ideally, more data should benefit downstream performance, but the risk of catastrophic forgetting is also greater due to training the model with a larger dataset on a specific task.

To test how our Multi-brain-tuning affects downstream performance, we test how downstream performance scales with the data size used for brain-tuning. We do this for 2 tasks (namely Phonemes Prediction and Phonetic Sentence Type prediction), following Moussa et al. [2025]. Detailed formulations of these tasks can be found in Supp.A.4. We use linear probes to perform the tasks and report the mean over model layers. In addition to the Multi-brain-tuned model, we also report the Single-brain-tuned and the LLM-tuned models for comparison.

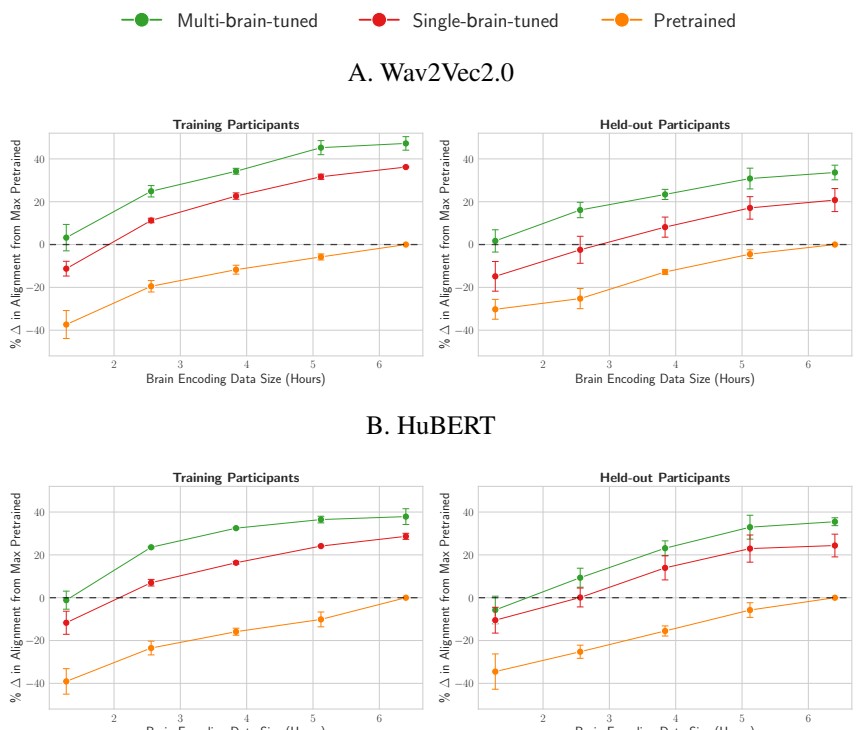

Figure 2: Brain encoding efficiency for both training and held-out participants and two model families (Wav2Vec2.0 and HuBERT). Brain-tuned models consistently outperform pretrained counterparts on both training and held-out participants. Multi-brain-tuned models consistently outperform Single-brain-tuned ones, reaching the max. pretrained performance with approximately a fifth of the brain encoding data and improving brain alignment up to $50\%$ from the max. pretrained brain alignment.

## 4 Results

### 4.1 Brain Alignment Efficiency

We evaluate Brain Alignment Efficiency by varying the size of brain encoding data as detailed in Sec.3.4.2. This enables us to measure the fraction of encoding data required by a tuned model to achieve or surpass the best brain alignment achieved by the pretrained model, which is obtained using the full encoding data. We report this in Fig.2 across two model families (Wav2Vec2.0 and HuBERT) for the training participants (seen during Brain-tuning) and the held-out participants (unseen by all models). We note that all brain alignment is evaluated on fMRI data held-out from training.

For the training participants (left plots of Fig.2), both Multi-brain-tuned (Sec.3.3.2) and the Single-brain-tuned (Sec.3.3.4) models match the maximum pretrained performance using approximately one-fifth of the data. With more encoding data, both models continue to improve, reaching up to $50\%$ more alignment when using the full encoding data. Notably, the Multi-brain-tuned model consistently outperforms the Single-brain-tuned model across all evaluated data fractions, alleviating the need for the standard practice of building Single-participant models to attain the best performance.

The Multi-brain-tuned model maintains its efficiency advantage for held-out participants (right plots of Fig.2), reaching the pretrained performance with roughly one-fifth of the data, whereas the Single-brain-tuned model requires approximately twice this amount. Both models continue to improve with more data, with a clear advantage for the Multi-brain-tuned model. These results are consistent across both the Wav2Vec2.0 and HuBERT model families. The comparable efficiency of the Multi-brain-tuned model on training and held-out participants strongly suggests that Multi-brain-tuning leads to more efficient and general representations that are beneficial for unseen participants.

A. Performance scaling with tuning data size

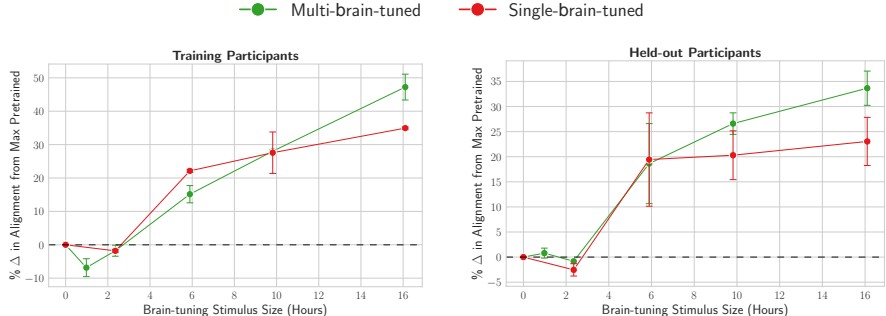

B. Flat map visualization of improvement over pretrained        C. Improvement on Narratives

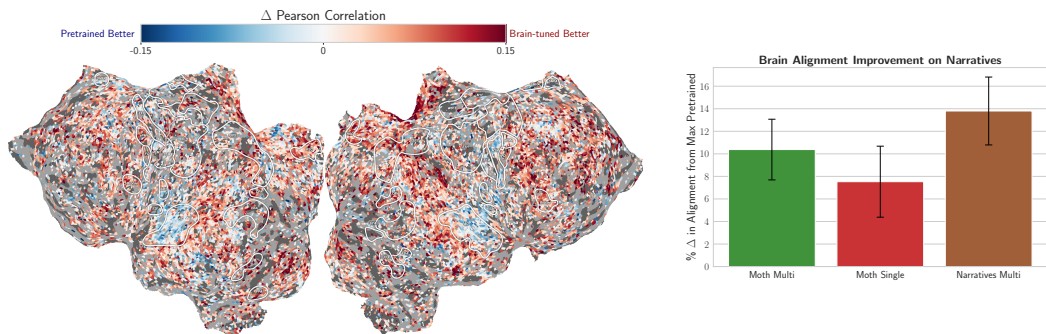

Figure 3: Brain-tuning generalization. (A) Scaling with tuning data size, showing that Multi-brain-tuning scales better, especially on held-out participants. (B) Voxel-wise change in brain alignment after Multi-brain-tuning for a held-out participant from the Moth Radio hour dataset (the remaining participants are reported in Supp.B.2), showing improved alignment across frontal and parietal regions. (C) Alignment improvement on novel stimuli and participants (from the Narratives dataset), indicating the capability of Multi-brain-tuned models for cross-dataset generalization.

## 4.2    Brain-tuning Generalization

In Sec.4.1, we demonstrated that Multi-brain-tuning enhances alignment magnitude and efficiency for training and held-out participants. Here, we investigate how the brain-tuned models generalize when provided with increasing amounts of data during brain-tuning and how these models perform on a diverse out-of-distribution dataset.

Fig.3A reports the change in brain alignment (relative to the pretrained Wav2Vec2.0) as a function of increasing stimulus set size. The results for HuBERT can be found in Supp.C.1. Generally, Multi-brain-tuning strongly benefits from increasing the tuning data size. For training participants, both Multi- and Single-brain-tuned models demonstrate increased alignment as the tuning data grows, although the Multi-brain-tuned model consistently performs better when the entire stimulus set is utilized. As for the held-out participants, both models perform similarly when the tuning data size is less than 6 hours. Nonetheless, Multi-brain-tuned models show a strong upward trend when more tuning data is used, while the Single-brain-tuned ones saturate. Overall, Multi-brain-tuning shows a greater ability to improve and generalize with more brain-tuning training data.

When we visualize the improvement in brain alignment of the Multi-brain-tuned over the pretrained Wav2Vec2.0 for held-out participants from the Moth Radio Hour dataset (Fig.3B, Supp.B.2), we observe a widespread improvement across the brain, especially in the frontal and parietal regions. The auditory cortex shows a slight decrease in alignment. This is because we report alignment over the upper-middle and later layers of the models, which encode more semantics. An additional reason

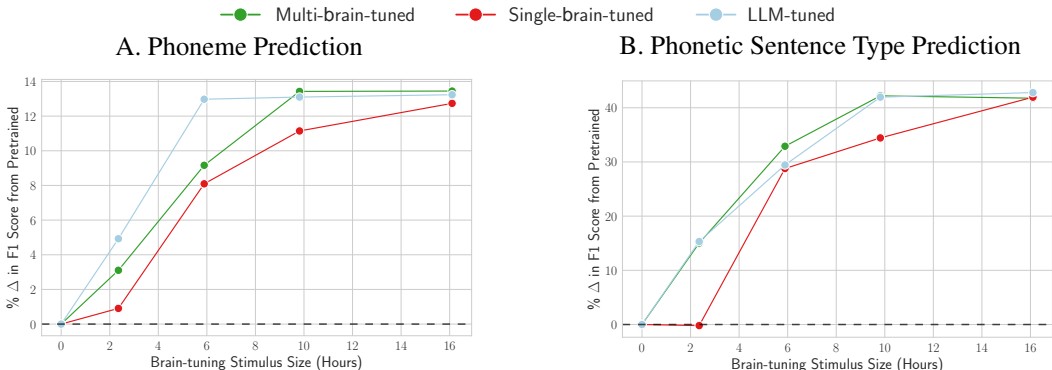

Figure 4: Scaling of downstream performance with tuning data size. Brain-tuned models' performance increases with more data, with the Multi-brain-tuned taking less data to match the LLM-tuned model.

may be that the auditory cortex can be dominated by the larger semantic language areas during brain-tuning. This explanation is also supported by the results of [Vattikonda et al., 2025].

Finally, Fig.3C shows the mean brain alignment improvement in late language regions for Multi- and Single-brain-tuned models, which are tuned on the Moth Radio Hour dataset, when evaluated on an entirely new dataset – 16 participants of the Narratives dataset. Both Moth-Brain-tuned models show improvement over their pretrained ccounterparts, with the Moth-Multi-brain-tuned being considerably better. In fact, Moth-Multi-brain-tuned does not lag much behind the Narratives-Multi-brain-tuned (i.e., a model tuned using the same 16 participants from the Narratives dataset), which indicates a strong generalization ability to novel stimuli and participants.

Our results confirm the scalability of Multi-brain-tuning in improving alignment with unseen participants and stimuli when trained on sufficient amounts of data. The strong upward trend also indicates room for further improvement if more data is integrated during brain-tuning. We further corroborate these findings by showing that this scaling trend cannot be achieved with stimulus-tuning or LLM-tuning in Supp.B.3. Next, we investigate the downstream performance of brain-tuning.

### 4.3 Downstream Performance

While the gains of Multi-brain-tuning in brain alignment are clear (Sec.4.2 and 4.1), it's important to verify that Multi-brain-tuning doesn't lead to catastrophic forgetting or degrade downstream performance. Here, we evaluate the downstream performance of the tasks described in Sec.3.4.4 as a function of tuning data size. We vary the size of the data similarly to Sec.4.2 (i.e., by increasing the size of the brain-tuning training stimulus set). In addition to reporting the performance of Single-brain-tuned models, we also include the LLM-tuned baseline as it was shown to substantially improve performance on similar tasks by Moussa et al. [2025] and Vattikonda et al. [2025].

Fig.4 shows the percent improvement over the pretrained Wav2Vec2.0 model. Similar results for HuBERT can be found in Supp.C.2. Across all data fractions, brain-tuned models never underperform the pretrained model, ruling out catastrophic forgetting. Impressively, the Multi-brain-tuned model matches the LLM-tuned performance when the size of the data increases. Both Multi- and Single-brain-tuned models benefit strongly from more tuning data, with the Multi-brain-tuned performing better at smaller tuning data sizes. Overall, Multi-brain-tuning improves downstream performance, which also scales up with the amount of data, eventually matching the LLM-tuned baseline.

### 4.4 Effect of Tuning Objective and Model Size

In the previous sections, we reported the results using the $L_2$ objective and LoRA rank-8 updates. Here, we explain the reasons for these choices by comparing the Multi-brain-tuning behavior for different LoRA ranks and training objectives on Wav2Vec2.0 (Fig.5).

Fig.5A reports gains in brain alignment at several brain-tuning training data sizes for different LoRA ranks. We can observe that increasing the LoRA rank beyond 8 doesn't improve performance,

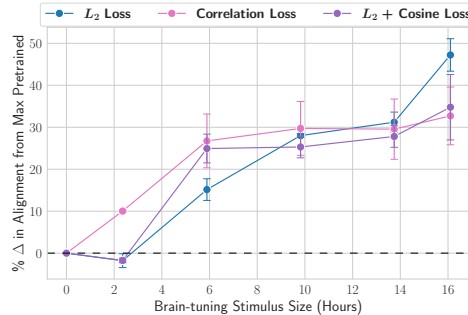

Figure 5: Effect of varying the loss function and LoRA rank on Multi-brain-tuning with (A) showing that Rank-8 updates perform best, and (B) indicating that the $L_2$ loss scales better.

especially when the data size increases. We find that even larger LoRA ranks as well as brain-tuning the full model do not improve performance over using rank-8 updates (see Supp.B.1).

Fig.5B contrasts different training losses for increasing tuning data sizes. Generally, $L_2$ performs better when the size of the brain-tuning training data increases, while the other two losses saturate. At low brain-tuning data sizes ($\leq 6$ hours), the Correlation loss outperforms other losses. While these results motivate using the $L_2$ loss because it scales better, they also highlight that the brain-tuning loss can have a large effect on brain-tuning. Future work may find improved brain-tuning losses.

## 5 Discussion and Conclusion

In this work, we introduced a novel brain-tuning approach that significantly improves the generalizability and efficiency of brain alignment in pretrained speech models. By fine-tuning language models on brain responses from multiple participants exposed to speech stimuli, we demonstrated that our method effectively addresses the limitations of existing participant-dependent approaches. Our extensive experiments showed that brain-tuning not only reduces the fMRI data requirements by a factor of five but also increases brain alignment by up to 50% and generalizes to new brain datasets.

These results highlight the potential of brain-tuning to create robust, participant-agnostic models that generalize well across individuals, paving the way for more scalable and inclusive approaches to studying language processing in the brain. Moreover, our comprehensive ablation studies on training loss and model parameters establish best practices for implementing brain-tuning in speech models.

Notably, brain-tuning also improves speech models' downstream performance on semantic tasks. This finding suggests that incorporating brain data during fine-tuning not only aligns models with the human brain in efficient and generalizable ways but also leads to representations that better capture generalizable semantic information. This direct evidence for a bidirectional benefit between neuroscience and AI contributes towards bridging the gap between the two fields.

**Limitations and Future Work.** We focused here on language-related brain regions, as they are most directly involved in processing the naturalistic speech stimuli used in our experiments. Future work can leverage our brain-tuning method for non-language regions or specific brain areas to gain new insights into their functional roles. Notably, our method is flexible and can be adapted to target different brain regions. Second, our experiments were conducted exclusively in English, reflecting the availability of large public fMRI datasets. Investigating brain-tuning with multilingual data in the future can assess whether brain-tuning can learn language-independent, generalizable semantic representations. Lastly, while we explored several training losses, there remains potential for developing new loss functions that lead to even better data efficiency and generalization.

By making our code and trained models publicly available, we aim to foster reproducibility and encourage further research on brain-tuning. We hope that our work will not only advance the integration of language models into cognitive neuroscience but also inspire new approaches to modeling human language comprehension.

## Acknowledgments

This work was partially funded by the German Research Foundation (DFG) - DFG Research Unit FOR 5368 and by the Max Planck Institute for Software Systems.

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

# A  Additional Methodology Details

## A.1  Brain ROIs Details

The Glasser Atlas for human cerebral cortex parcellation has 180 labeled ROIs per hemisphere [Glasser et al., 2016]. From these labels, we extract the following regions to be used during brain-tuning: Angular gyrus, lateral temporal cortex, inferior frontal gyrus, and middle frontal gyrus [Oota et al., 2024, Desai et al., 2023]. It also has the primary auditory and the early auditory regions. Fig.6 highlights the ROIs used for brain-tuning on the right hemisphere. Table 1 details each region and the ROI labels that cover it from the parcellation atlas.

Table 1: Brain regions and corresponding ROI labels.

| Region | Labels |
|---|---|
| Angular gyrus (AG) | PFm, PGs, PGi, TPOJ2, TPOJ3 |
| Lateral temporal cortex (LTC) | STSda, STSva, STGa, TE1a, TE2a, TGv, TGd, A5, STSdp, STSvp, PSL, STV, TPOJ1 |
| Inferior frontal gyrus (IFG) | 44, 45, IFJa, IFSp |
| Middle frontal gyrus (MFG) | 55b |
| Primary auditory cortex (A1) | A1 |
| Early auditory regions | A1, PBelt, MBelt, LBelt, RI, A4 |

## A.2  Noise Ceiling Calculation

Noise in fMRI data is very common and can impair brain-tuning and brain alignment estimation, so it is important to estimate the noise ceiling of each voxel in the fMRI responses. The voxel-wise noise ceiling is estimated for all participants based on the method by the fMRI dataset paper LeBel et al. [2024]. This method leverages repetitions of the same story for the participant (e.g., a story is repeated 10 times), then uses these repetitions to compute the maximum explainable variance for each voxel. This noise ceiling value estimates the amount of explainable variance in the brain signal, ranging from 0 to 1. We use this estimated noise ceiling to normalize the brain alignment during brain alignment estimation, as mentioned in Sec.3.4.1 of the main paper.

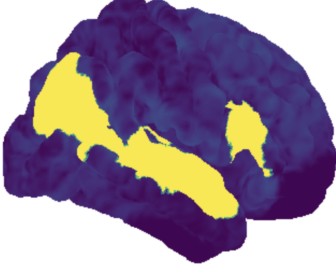

Figure 6: Brain-tuning ROIs. Yellow-highlighted regions are used for brain-tuning.

## A.3  Loss Functions and Training Details

Here, we detail the formulations of the different loss functions compared in Sections 3.3 and 4.4 of the main paper. We then compare different training techniques and alternatives for Multi-brain-tuning, showing that the one we used in Sec.3.3.2 performs best.

### A.3.1 Loss Functions

We define the loss functions over a batch $B$ of audio-fMRI pairs for Participant $i$, where the ground-truth fMRI responses are $R^i$ and the predicted fMRI responses are $\hat{R}^i$.

$L_2$ **Loss**. We compute the $L_2$ loss between $R^i$ and $\hat{R}^i$ as follows:

$$\mathcal{L}_{l2} = \frac{1}{|B|} \sum_{b=0}^{|B|-1} (R_b^i - \hat{R}_b^i)^2 \tag{1}$$

**Correlation Loss**. We compute the correlation loss between $R^i$ and $\hat{R}^i$ as follows (where corr is the correlation over voxels):

$$\mathcal{L}_{\text{corr}} = \frac{1}{|B|} \sum_{b=0}^{|B|-1} (1 - \text{corr}(R_b^i, \hat{R}_b^i)) \tag{2}$$

**Cosine + $L_2$ Loss**. We compute the Cosine + $L_2$ loss between $R^i$ and $\hat{R}^i$ as follows (where cos is the cosine similarity over voxels):

$$\mathcal{L}_{\text{cos}} = \frac{1}{|B|} \sum_{b=0}^{|B|-1} (1 - \cos(R_b^i, \hat{R}_b^i)) \tag{3}$$

Then we use it alongside the $L_2$ Loss (with $\lambda = 0.5$):

$$\mathcal{L}_{\text{cos-l2}} = \mathcal{L}_{\text{cos}} + \lambda \mathcal{L}_{l2} \tag{4}$$

### A.3.2 Comparing Multi-brain-tuning Techniques

The training strategy we use in the paper is predicting each participant's FreeSurfer ROIs responses independently (but using the same projection head), then computing the loss, and updating the model parameters. We detail here other techniques and alternatives and compare them to this setting, highlighting that our approach works best.

**Loss Average**. In the Loss average method, we compute the loss for each participant, then average it across participants before updating the parameters.

**Response Average**. For the Response average method, we average responses over participants, then compute the loss and update the parameters.

**Separate Heads**. Another alternative is to use a separate projection head for each participant. While this might work well with a limited number of participants, it will increase the training parameters considerably when fine-tuning with many participants.

**SRM-tuned**. Instead of FreeSurfer, we could potentially use other multi-participant alignment methods (e.g., SRM: Shared Response Modeling [Chen et al., 2015]) and apply the same tuning method. One limitation of SRM is the reduced projection dimension and difficulty in controlling the included ROIs.

**Non-linear Heads**. Lastly, rather than one FC layer to predict brain responses from the output tokens, we could use a non-linear network (of 2 or more fully connected layers).

Fig.7 reports the brain alignment improvement over the pretrained Wav2Vec2.0 for all aforementioned alternatives. It shows that our method works better than these alternatives as it allows the models to learn more robustly from information across participants.

### A.4 Downstream Tasks

We elaborate here on the datasets and the formulation of the downstream tasks mentioned in Sections 3.4.4 and 4.3 of the main paper.

**Phonemes Prediction**. Phoneme recognition is done as a multi-label classification problem, following the work of Moussa et al. [2025]. A linear classifier projects the layer representation to a set of 39 possible phonemes that occurred in the original input audio segment. We use the TIMIT dataset

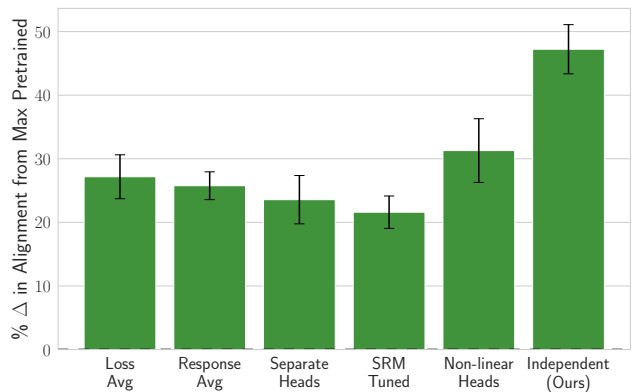

Figure 7: Comparing Multi-brain-tuning using FreeSurfer projection and independent loss computations(i.e., the loss is computed and backpropagated for each participant separately) versus other alternative techniques (refer to Sec.3.3.2 of the main paper).

[Garofolo, 1993] because of its phonetically rich audio snippets. The final performance measure is the classifier's F1-score on the held-out test set. We report the F1-score averaged over the upper-middle layers as done for brain alignment (refer to Sec.3.4.1 in the main paper).

**Phonetic Sentence Type Prediction**. Predicting the phonetic sentence type can be used to evaluate a model's phonetic understanding beyond single phonemes (or words). The TIMIT dataset [Garofolo, 1993] has one of three phonetic types for each utterance: SA (for utterances that cover all English phonemes), SX (for phonetically balanced utterances that cover many phones with few words), and SI (for natural and phonetically diverse utterances). Each of the three types (SA, SX, SI) highlights specific speech dialectal or phonetic aspects. To evaluate performance on this task, we follow [Moussa et al., 2025] to predict the phonetic sentence type. We add a projection classification head to predict the sentence type from the given layer's representation. The performance is measured by the F1-score on the held-out test set, averaged across the upper-middle layers.

# B    Additional Results

## B.1    Extended LoRA Rank Ablations

Fig.8 extends Fig.5 by adding Rank-32 updates and all model updates (fine-tuning all transformer parameters). It supports our finding that we don't need more than rank-8 updates for our method to work well. Moreover, it shows that updating the entire model scales more slowly than LoRA, indicating that it needs more data to reach the same performance.

## B.2    Additional Brain Alignment Plots

Here, we visualize the impact of brain-tuning on brain alignment for the remaining held-out participants. Fig.12 extends Fig.3B by showing the remaining 4 participants. Similarly to Fig.3B, we observe a widespread improvement across the brain for these participants, especially the frontal and parietal regions, while the auditory cortex shows a slight decrease in alignment. We attribute this decrease in auditory cortex alignment to the fact that we report alignment over the upper-middle and later layers of the models, which are known to be more semantic (refer to Sec.4.2 in the main paper for more details).

## B.3    Brain Alignment Generalization of LLM-tuning and Stimulus-tuning

In this section, we elaborate on the training details of LLM-tuning and Stimulus-tuning, then report on how they scale with more tuning data.

**LLM-tuning**. For tuning, we use representations from layers 18 to 24 of the Llama2-7B Model [Touvron et al., 2023] instead of brain signals. These layers are used because they show the best

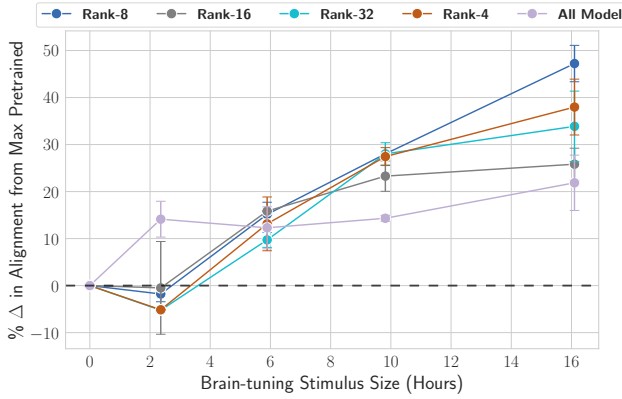

Figure 8: Effect of the number of trainable parameters for Wav2Vec2.0 (Extending Sec.4.4 of the main paper). Increasing the number of trainable parameters with a higher rank (e.g., 32) or by fine-tuning the entire model doesn't lead to better scaling than rank-8 updates.

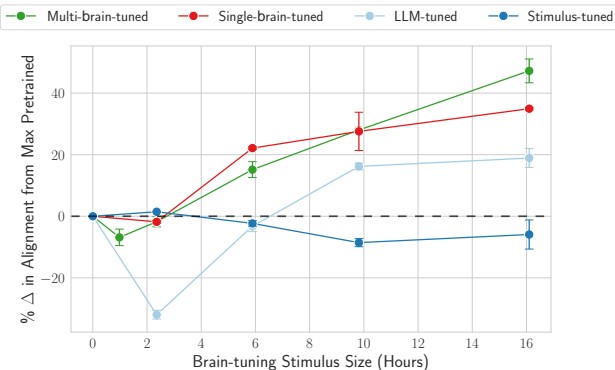

Figure 9: Brain Alignment scaling with LLM-tuning and Stimulus-tuning of Wav2Vec2.0. This figure extends Sec.4.2 of the main paper by reporting the improvement in alignment with more data for LLM-tuning and Stimulus-tuning. It shows that LLM-tuning can help improve brain alignment with more data, but it tends to saturate and is always worse than brain-tuning. Stimulus-tuning doesn't seem to improve alignment and always performs comparably to the pretrained model.

alignment with late language regions. We then apply a similar fine-tuning pipeline to that of brain-tuning (detailed in Sec.3.3.3 of the main paper). We use LoRA rank-8 updates, a learning rate of $10^{-4}$ with linear decay, and a batch size of 128 samples of (audio, fMRI response) pairs. For LLM-tuning, we found that it takes longer to converge than brain-tuning; we train them for 250 epochs, which takes around 10h on two NVIDIA A40 48GB GPUs.

**Stimulus-tuning**. This baseline aims to test the benefits of brain-tuning against simply fine-tuning using stimulus audio. This highlights any improvements in the model that would be solely due to seeing more data. We follow the training setting in [Moussa et al., 2025] for stimulus-tuning. The same pretraining losses (the diversity loss and the contrastive loss) with the same hyperparameters of [Baevski et al., 2020] are used. The model is then fine-tuned for 300 epochs using a base learning rate of $2 \times 10^{-5}$ with a warm-up for the first $10\%$ of the updates, followed by a linear decay schedule. It takes around the same amount of time to train as LLM-tuning.

Next, we test whether scaling the data for LLM-tuning and Stimulus-tuning leads to improved alignment, as we observe with brain-tuning (refer to Sec.4.2). Fig.9 shows the change in brain-alignment for brain-tuning as well as LLM-tuning and Stimulus-tuning on the training participants. LLM-tuning improves alignment with more data, but it shows a saturation trend and is always lower than brain-tuned models. Stimulus-tuned models don't show improvement over the pretrained counterpart, indicating that seeing more audio data is not the cause for the improved alignment.

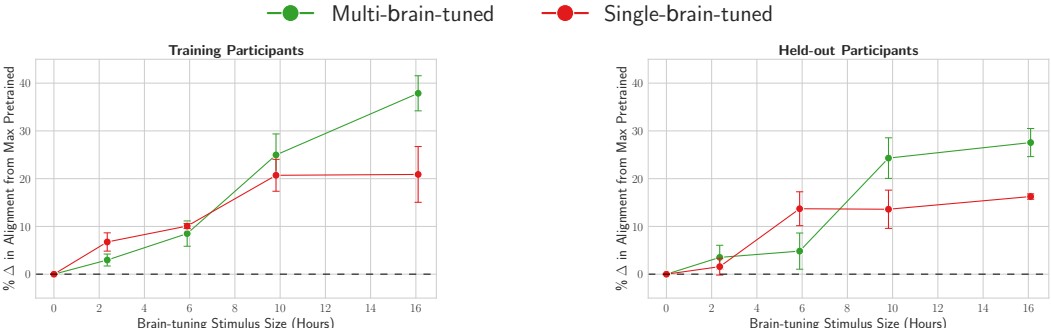

Figure 10: Impact of scaling the tuning data of brain-tuning on brain alignment for HuBERT. Similar to brain-tuned models of Wav2Vec2.0 (Sec.4.2), when the tuning data scales up, Multi-brain-tuned models perform better than Single-brain-tuned ones, on both training and held-out participants.

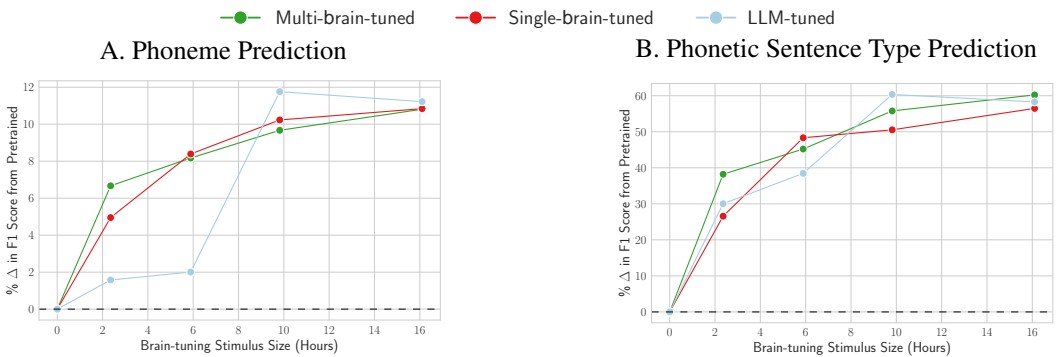

Figure 11: Scaling of downstream performance with tuning data size for HuBERT. Brain-tuned models' performance increases with more data, eventually matching that of the LLM-tuned model.

## C   HuBERT Results

### C.1   Generalization Results

We repeat here the same analysis in Sec.4.2 but for the HuBERT model family. We test how the brain-tuned models generalize against increasing amounts of data during brain-tuning. This is done by measuring brain alignment improvement (relative to pretrained HuBERT) when we scale the data used for Multi- and Single-brain-tuning. Fig.10 shows a similar trend to Fig.3 on both training and held-out participants. When we increase the data used for brain-tuning, Multi-brain-tuned models tend to perform better than Single-brain-tuned ones. As for lower data fractions, the improvement of both was comparable. These results (along with their Wav2Vec2.0 parallels in Sec.4.2 of the main paper) further confirm the scalability of Multi-brain-tuning in improving alignment with new unseen participants. The upward trend also indicates the potential for further improvement if more data is integrated for brain-tuning.

### C.2   Downstream Results

We report here the downstream performance of HuBERT on the same tasks detailed in Sec.4.3 and Supp.A.4. Fig.11 shows similar findings to Fig.4 of the main paper. When the amount of tuning data increases, brain-tuned models eventually reach the same level of performance as the LLM-tuned one (which was shown to substantially improve performance on similar tasks by [Moussa et al., 2025, Vattikonda et al., 2025]). Moreover, for all data sizes, brain-tuned models never perform worse than their pretrained counterparts. These results (alongside their Wav2Vec2.0 equivalents in Sec.4.3 of the main paper) further confirm that our brain-tuning approach doesn't lead to catastrophic forgetting; on the contrary, it leads to a strong improvement in downstream performance.

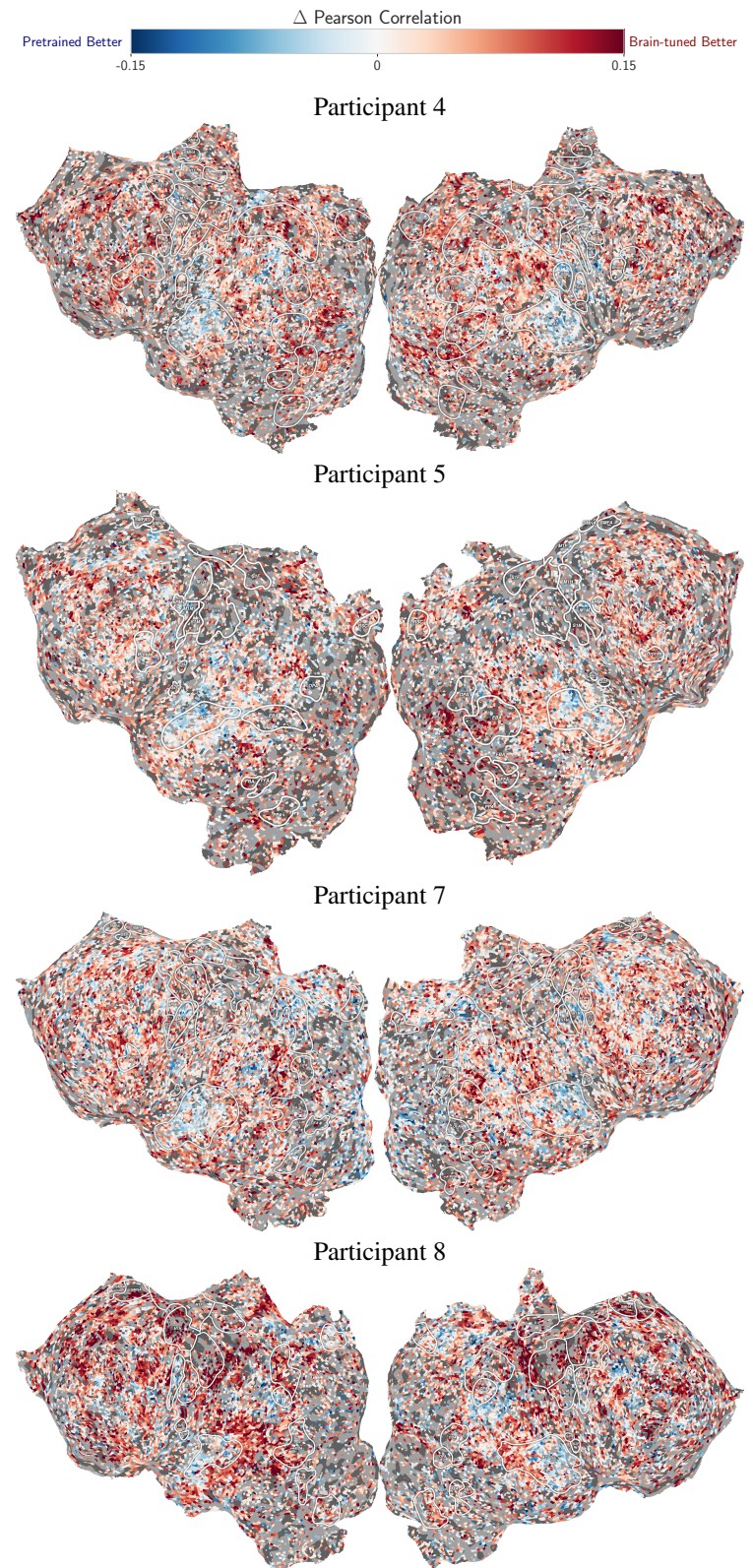

Figure 12: Impact of Multi-brain-tuning on brain alignment for held-out participants. The figure shows the change in brain alignment (measured by Pearson Correlation) after Multi-brain-tuning, compared to the pretrained Wav2Vec2.0 model. It shows a widespread improvement across the brain for these participants, especially the frontal and parietal regions.

