# OpenReview forum: "Brain-tuning Improves Generalizability and Efficiency of Brain Alignment in Speech Models"
_NeurIPS.cc/2025/Conference — NeurIPS 2025 poster_

### Official Review · Reviewer_W1Kc · 2025-06-26

**Clarity:** 3
**Significance:** 2
**Originality:** 1
**Rating:** 4
**Confidence:** 3

**Summary:**

This paper proses a novel approach to improve model prediction accuracy of naturalistic fMRI data. This approach leads to two notable improvements over more traditional methods. The first is a decrease in the amount of fMRI data for new participants and the second is an increase in brain alignment. The authors accomplish them by using 2 different pre-trained speech models (wav2vec2 and HuBert) and fine tuning these models on (1) the individual participants data by adding a layer onto the model (2) on standardized data across participants.

**Questions:**

1. How do the advances of this paper compare to Tang & Huth and BrainWavLM? I think it would be beneficial for the author to make it clearer why this verison of the method is significant.

2. Have the authors spent any time interpreting the weights from the aligned model? Is it more interpretable than other models? It seems likely that this will be a harder architecture to interpret which will limit its significance and usability in many/most research settings?

**Ethical Concerns:**

["NO or VERY MINOR ethics concerns only"]

**Final Justification:**

I think this is a technically solid paper that lacks novelty. However given most reviewers seem to recommend accept and that the paper does appear to be technically solid, I have chosen to raise my recommendation to a borderline accept. The authors have also done a good job integrating and addressing comments from reviewers.

**Limitations:**

yes

**Quality:**

3

**Strengths And Weaknesses:**

The submission is technically sound and robust. The quality and clarity of this work is definitely it strongest point. However the originality and significance of this work is reduced by both recent publications in other outlets and the limited scholarship of the work. This papers primary results seem to highly overlap with Tang & Huth 2025. Which while a relatively new work shows many of the same results that aligning participants data decreases the amount of data required per person. The tang paper used language models instead of speech models. Similarly this paper is highly overlapping with BrainWavLM (Vattikonda et al, 2025). The primary difference being the generalizability across participants. However both of these improvements seem relatively trivial from previous work in the field.

Notably, the paper heavily cites a single author's previous work, while omitting relevant related research from other groups. This raises concerns about scholarly balance and the framing of the work within the broader field. A more thorough literature review that situates this contribution among alternative or complementary approaches would strengthen the paper and make its novelty clearer to readers outside the authors’ own research lineage. This work shows a strong bias towards ML papers in its citations over neuroscience papers, which will limit its reach. In particular this work fails to cite Tang & Huth 2025 (as mentioned above) which shows similar decreases in the amount of fMRI data needed to predict brain data and an increase in robustness when aligning across participants. Thus the primary results of this paper have been shown already in the literature with another method. The authors do not at any point address this redundancy or highlight differences. Similarly the authors claim that accurately estimating brain alignment has been shown to require a large amount of per-participant data (line 66-67) when this has been shown in work by authors other than Antonello first (e.g. LeBel et al 2023, Lebel et al 2021).

As a small note, the authors refer to LMs as "model organisms" (line 24). "model organisms" is already a standard term in the field referring to biological organisms (i.e. flies or mice). Using it here to suggest LMs is likely to be misleading. The term "models" is likely clearer and more accurate.

---

> ### Author Rebuttal · Authors · 2025-07-30
>
> We thank the reviewer for their review of our work and literature suggestions. Below, we clarify the distinctions of our contributions compared to the mentioned previous work (further supporting it with new results), address interpretability concerns, and discuss the suggested relevant literature.
>
> ## **Q1: Novelty Compared to Recent Work**
>
>
> ### **A. Distinction from Brain decoding work by Tang & Huth (2025)**
>
> We appreciate the pointer to the recent decoding work by Tang & Huth (2025). However, we want to highlight that this study investigates generalization in brain decoding rather than brain encoding (and is not the first to show decoding generalization). The two domains differ a lot, and transferring methods between them can be challenging.
>
> Tang & Huth (2025) demonstrate strong generalization across stimulus modalities, extending previous decoding successes in speech [1], language [2, 3, 4], and vision [5, 6, 7] in unifying participant responses and generalizing to new participants. Some of these methods show impressive generalization [8, 9], including zero-shot decoding [10]. Despite these advances, achieving similar generalization in brain encoding remains **challenging and unresolved**. Previous attempts typically resulted in substantial performance drops compared to single-subject models (see below). Adapting decoding approaches directly may not be straightforward or even feasible for encoding tasks, as discussed in [11]. This emphasizes the non-triviality of generalization in brain encoding,  an important gap that our work addresses.
>
> ### **B. Challenges in Brain Encoding Generalization**
>
> Unlike decoding, brain encoding models typically fail to generalize effectively to new participants. Common participant-alignment methods, such as SRM[22] and FreeSurfer, substantially degrade encoding performance (**~40%** drop, as shown below; cross-participant encoding models trained on 3 participants and tested on a 4th). Even recent fine-tuning approaches like BrainWavLM [15] show around a **30%** drop in cross-participant performance (where the speech model is fine-tuned on one participant, but the encoding model is trained and evaluated on others).
>
> | **Method** | **Single participant** | **Cross participant** | **Avg performance drop** |
> | --- | --- | --- | --- |
> | SRM (brain alignment) | 0.104 ± 0.013 | 0.059 ± 0.012 | 42.5% |
> | FreeSurfer (brain alignment) | 0.12 ± 0.021 | 0.072 ± 0.009 | 40.2% |
>
> Our approach is, to our knowledge, the first to demonstrate significant improvements and generalization in speech and language encoding by fine-tuning with neural data. Crucially, our method simultaneously improves performance on both trained and held-out participants, addressing an important gap unresolved by prior work.
>
> ### **C. Comparison with BrainWavLM (Vattikonda et al., 2025)**
>
> Both Brain-tuning [14] and BrainWavLM [15] introduce a framework for fine-tuning speech models with fMRI data, with Brain-tuning [14] being the first such work. However, neither paper offers a way to adapt to multi-subject or multi-dataset settings; they also show limited transferability across subjects. We emphasize that successful generalization across participants requires extensive, careful design choices and **is not a trivial extension of those works**. Specifically, our results demonstrate:
>
> 1. **The method of aggregating and presenting multiple participant responses** critically influences generalization (Fig. 6, Supp A.3).
> 2. **Loss function and model capacity** substantially impact performance and generalization ability (Fig. 4).
> 3. ***new results,* Investigating different methods for Multiple participant handling**: In order to further demonstrate the effectiveness of our design choices, we explore the performance of different multiple participants handling methods. Specifically, we compare our Multi-brain-tuning against three alternatives: (i) separate participant-specific projection heads (as in prior work [23]), (ii) a non-linear projection network (MLP instead of linear), and (iii) SRM-based participant alignment (instead of FreeSurfer). The table below summarizes alignment improvements for training and held-out participants.
>
> |  | **Our Multi-brain-tuned** | **With Separate Heads** | **With SRM** | **With Non-linear Projection** |
> | --- | --- | --- | --- | --- |
> | Train (mean ± STE) | **47.22 ± 3.15** | 23.57 ± 3.8 | 21.6 ± 2.55 | 31.29 ± 5.01 |
> | Held-out (mean ± STE) | **33.66 ± 3.41** | 23.2 ± 6.76 | 9.8 ± 1.6 | 30.12 ± 5.73 |
>
> Our Multi-brain-tuned model outperforms all alternatives while being:
>
> - more parameter-efficient compared to separate heads or non-linear alternatives, which scale poorly with participant number (e.g., 16 separate projection heads for Narratives).
> - more controllable in selecting targeted brain areas unlike SRM.
> - easier to apply across multiple datasets due to FreeSurfer’s widespread adoption.
>
> ## **Q2: Interpretability of Multi-Brain-Tuned Models**
>
> We argue that our multi-brain-tuned models are **at least as interpretable** as their pretrained counterparts. Our method fine-tunes the full architecture, unlike BrainWavLM [15], which removes some of the model layers. Thus, existing interpretability analyses for speech models (layer-probing, attribution, representation analysis) remain fully applicable.
>
> Moreover, understanding and interpreting why models align with the brain in the first place is an active research area ([16, 18, 19, 20, 21]. Methods developed in this context apply equally to our Multi-brain-tuned models, potentially yielding further insights into the effect of increased neural alignment. For instance, recent work [16] demonstrates that speech brain-tuned models’ layers reflect hierarchical brain processing stages better than pretrained models. Similar analyses can be easily employed on our Multi-brain-tuned models, and we hope that our work inspires such future work.
>
> In essence, since our models can be used, analyzed, and deployed in the same manner as their popular pretrained counterparts, our approach wouldn’t complicate interpretability analyses of these models.
>
> ## **Q3: Revising Literature and Terminology**
>
> We appreciate the reviewer’s feedback regarding scholarly literature balance. We note, nevertheless, that our related work discussion covers relevant brain encoding work, and that our repeated comparison to [14, 15] is because they are the only recent works related to brain-tuning of speech models. However, we agree that expanding our literature review to integrate more brain decoding and neuroscience-focused work (such as Tang & Huth, 2025, and others mentioned earlier) will better contextualize our contributions.
>
> Regarding terminology, our use of "model organisms" aimed at highlighting parallels with biological research models, distinguishing our usage from general AI "models." This term has been similarly employed in recent survey literature [17]. Nevertheless, we understand the potential confusion and will clarify this usage in the manuscript.
>
> We greatly appreciate the reviewer’s feedback and believe these clarifications significantly strengthen the paper. We hope these revisions clearly highlight the contributions, novelty, and interpretability of the work, and hope these improvements are reflected positively in your evaluation.
>
> ## **References**
>
> [1] Défossez et al. (2023). Decoding speech perception from non-invasive brain recordings. *Nature Machine Intelligence*.
>
> [2] Lévy et al. (2025). Brain-to-Text Decoding: A Non-invasive Approach via Typing.
>
> [3] d’Ascoli et al. (2024). Decoding individual words from non-invasive brain recordings across 723 participants.
>
> [4] Liu et al. (2025). Learning interpretable representations leads to semantically faithful EEG-to-text generation.
>
> [5] Benchetrit et al. (2024). Brain decoding: toward real-time reconstruction of visual perception. *ICLR*.
>
> [6] Liu et al. (2023). BrainCLIP: Bridging brain and visual-linguistic representation via CLIP for generic natural visual stimulus decoding.
>
> [7] Chen & Wei (2024). Mind’s Eye: Image recognition by EEG via multimodal similarity-keeping contrastive learning.
>
> [8] Thual et al. (2023). Aligning brain functions boosts the decoding of visual semantics in novel subjects.
>
> [9] Dahan et al. (2025). SIM: Surface-based fMRI analysis for inter-subject multimodal decoding from movie-watching experiments. *ICLR*.
>
> [10] Aristimunha et al. (2025). EEG Foundation Challenge: From cross-task to cross-subject EEG decoding.
>
> [11] Beliy et al. (2024). The wisdom of a crowd of brains: A universal brain encoder.
>
> [12] Schwartz et al. (2019). Inducing brain-relevant bias in natural language processing models. *NeurIPS*.
>
> [13] Lu et al. (2024). Achieving more human brain-like vision via human EEG representational alignment.
>
> [14] Moussa et al. (2025). Improving semantic understanding in speech language models via brain-tuning. *ICLR*.
>
> [15] Vattikonda et al. (2025). BrainWavLM: Fine-tuning speech representations with brain responses to language.
>
> [16] Moussa & Toneva (2025). Brain-tuned speech models better reflect speech processing stages in the brain. *Interspeech*.
>
> [17] Oota et al. (2023). Deep Neural Networks and Brain Alignment: Brain Encoding and Decoding (Survey). *TMLR*.
>
> [18] Caucheteux & King (2022). Brains and algorithms partially converge in natural language processing. *Communications Biology*.
>
> [19] Merlin & Toneva (2022). Language models and brains align due to more than next-word prediction and word-level information.
>
> [20] Oota et al. (2023). Joint processing of linguistic properties in brains and language models.
>
> [21] Oota et al. (2024). Speech language models lack important brain-relevant semantics. *ACL*.
>
> [22] Chen et al. (2015). A Reduced-Dimension fMRI Shared Response Model. *NeurIPS*.
>
> [23] Jayalath et al. (2025). The Brain’s Bitter Lesson: Scaling Speech Decoding With Self-Supervised Learning. *ICML*.

---

> > ### Comment · Reviewer_W1Kc · 2025-08-01
> >
> > I appreciate the authors consideration of the review and update to their paper. I still remain unconvinced of the novelty of this approach. The authors claim that encoding and decoding generalization differ substantially and that generalizing across participants in the case of encoding is significantly harder, but I think the evidence to support that is week. Using Freesurfer or SRM brain alignment as a comparison point of this is a straw man argument. I think many of the tools used to generalize decoding would easily apply to an encoding framework (for example the Tang & Huth paper which in its methods first generalizes and encoding model and then inverts it to do decoding showing that encoding generalization is possible) and I imagine this is true for other methods/papers pointed out by reviewer KUd9.
> >
> > I think there are many strengths to this paper and the authors appear to have taken the comments seriously, however I think this paper still lacks in novelty.

---

> > > ### Author Response · Authors · 2025-08-03
> > > **We further clarify our contributions, particularly in light of new results comparing to the approach by Tang & Huth (2025).**
> > >
> > > We thank the reviewer for their continued engagement and the opportunity to clarify our contributions, particularly with **new results** comparing to existing approaches like Tang & Huth (2025).
> > >
> > > ## **Limitations of Existing Multi-subject Methods**
> > >
> > > Common multi-subject methods like SRM [1], Hyperalignment [2], or linear converters (as in Tang & Huth 2025) primarily align participant responses rather than directly improving the pretrained representations. Notably, Tang & Huth (2025) employ participant-specific linear converters, requiring separate converters for each participant pair and for participants to share the same stimuli, thus significantly limiting scalability to larger cohorts. This method is similar to Hyperalignment and SRM methods that learns **individual** projection matrices to a shared space.
> > >
> > > ## **Comparison to the approach of Tang & Huth (2025) in encoding**
> > >
> > > Our approach differs **fundamentally** by directly fine-tuning pretrained representations to enhance the representations' predictive power and generalization, without the need for participant-specific converters or shared-response retraining for new participants with limited data. Our approach **does not** require all participants to have experienced identical stimuli, unlike these methods.
> > >
> > > As for the performance comparison to the converter method used by Tang & Huth (2025), we provide below **results inspired by their setup** and compare them to our Multi-brain-tuning approach.
> > >
> > > We learn converters from 2 participants to the third using **70mins** of story data, then train encoding models on the full data (around 6 hours) for these 2 participants, ensemble their predictions, and the converters to predict the responses for the third participant on test data.
> > >
> > > Our baseline in this case is using pretrained representations to train an encoding model trained on the 70mins of story data and evaluated using the same test data. We also compare to the encoding model trained on the **same 70mins** of story data but using the representations of the **Multi-brain-tuned model**. Please note that the participants tested are held-out and were not used during brain-tuning.
> > >
> > > The table below shows the percent improvement in brain alignment over the pretrained model for both the converter approach and the Multi-brain-tuning approach.
> > >
> > > |  | **Converter approach** | **Our Multi-brain-tuned** |
> > > | --- | --- | --- |
> > > | Pct Improvement (mean ± STE) | 13.66 ± 5.11 | **22.73 ± 5.60** |
> > >
> > > Our Multi-brain-tuned model outperforms the converter approach, providing stronger generalization without participant-specific retraining. Additionally, the converter method limits scalability and does not surpass training with more data on individual participants (decoding accuracy with converters is around half of full training on the full data - Tang & Huth 2025, Fig. 2B).
> > >
> > > ## **Limitations of adapting popular decoding approaches, including Tang &Huth 2025**
> > >
> > > While decoding methods like those suggested by reviewer KUd9 and Tang & Huth (2025) provide valuable insights, their objectives and methods differ from encoding tasks. Decoding methods primarily align participants’ neural responses, usually to a low-dimensional representation, to feed to a pretrained model for decoding. This dependency restricts their scalability, cross-dataset applicability, and ability to **predict full brain responses**.  Methods like linear converters cannot generalize across datasets without shared stimuli, and are computationally expensive to scale (e.g., full-brain converter matrices would have ~100k x 100k parameters).
> > >
> > > Furthermore, we want to stress a more fundamental principle; the merit of our method is not only in being better than these methods, but it’s in approaching the problem differently, providing a **simple yet effective** approach (as noted by reviewer cQaF) for encoding models to generalize and be more efficient. This direction was recently introduced for speech models in [3]. Our approach differs in objective from [3], but it builds effectively on it to provide better performance as well as systematic analyses to reach the best framework to do Multi-brain-tuning, a framework that we hope would inspire more work to adapt this to other settings like multi-modal and vision encoding models.
> > >
> > > We hope that providing these results alongside comparisons to some of the suggested methods by reviewer kUd9 highlights the advantage of our approach better.  We think that this comparison to some of the popular decoding approaches would be a strong addition to the paper, and we thank the reviewer again for the suggestions.
> > >
> > > ## References
> > > [1] Chen et al. (2015). A Reduced-Dimension fMRI Shared Response Model. *NeurIPS*.
> > >
> > > [2] Hexby et al. (2020). Hyperalignment: Modeling shared information encoded in idiosyncratic cortical topographies. *Elife*.
> > >
> > > [3] Moussa et al. (2025). Improving semantic understanding in speech language models via brain-tuning. *ICLR*

---

### Official Review · Reviewer_KUd9 · 2025-06-29

**Clarity:** 3
**Significance:** 3
**Originality:** 2
**Rating:** 5
**Confidence:** 4

**Summary:**

The authors propose improving the data-efficiency and generalisation of brain-aligning LLMs with new subjects. They use speech perception fMRI data, transform this data into a unified representation-space across participants and propose training a projection layer and fine-tuning the pre-trained LLM with multiple new participants exposed to the same stimulus. The results provide support for these claims.

**Questions:**

Please see weaknesses. In general, this seems like interesting and valuable work, and I am glad to see the authors addressing this problem.

**Ethical Concerns:**

["NO or VERY MINOR ethics concerns only"]

**Final Justification:**

The authors have satisfied my main concern by conducting an experiment on an entirely new dataset with novel stimuli. They have also addressed my more minor concerns and questions about design choices for subject modelling in encoding models.

**Limitations:**

yes

**Quality:**

3

**Strengths And Weaknesses:**

Strengths:
- Authors tackle a highly prevalent problem in brain decoding/encoding---how to generalise to novel subjects.
- Well-designed experiments and evaluation methodology.
- The method is very simple, and seems to be effective.
- Inclusion of downstream generalisation experiments that verify alignment doesn’t degrade decoding.

Weaknesses:
- Subjects listen to the same natural stimuli, weakening the impact of generalisation from novel stimuli + novel subject generalisation to just novel subject generalisation. It would be valuable to see how the method performs on novel stimuli + novel subjects. Is this an experiment the authors could perform?
- Lines 118-120: Are varied projection dimensions really a problem? For example, [C] use conditional projections to put variable-dimension inputs into a shared representation space.
- The method of adding a projection head and LoRA fine-tuning the LLM is not particularly novel or insightful and has been done in similar subfields, e.g. [E]. Though the application to encoding models might be new.
- Minor: the related work could use more contextualisation. E.g. noting that brain decoding models (even if not encoding models) have previously achieved good subject generalisation (see [A, B, C, D, E, F]). Especially [D] with fMRI. Similarly, line 69-71 suggests prior methods unifying subjects haven’t focused on speech when [A, B, F] use subject-specific linear layers, [C] use FiLM-based subject conditioning, and [G] uses source space transformations to project into a shared space. All of these focus on speech decoding.

[A] Défossez, Alexandre, et al. "Decoding speech perception from non-invasive brain recordings." Nature Machine Intelligence 5.10 (2023): 1097-1107.

[B] d'Ascoli, Stéphane, et al. "Decoding individual words from non-invasive brain recordings across 723 participants." arXiv preprint arXiv:2412.17829 (2024).

[C] Jayalath, Dulhan, et al. "The Brain's Bitter Lesson: Scaling Speech Decoding With Self-Supervised Learning." ICML (2025).

[D] Tang, Jerry, and Alexander G. Huth. "Semantic language decoding across participants and stimulus modalities." Current Biology (2025).

[E] Yang, Yiqian, et al. "Neuspeech: Decode neural signal as speech." arXiv preprint arXiv:2403.01748 (2024).

[F] Jayalath, Dulhan, et al. "Unlocking non-invasive brain-to-text." arXiv preprint arXiv:2505.13446 (2025).

[G] Gideoni, Yonatan, et al. "Non-invasive Neural Decoding in Source Reconstructed Brain Space." arXiv preprint arXiv:2410.19838 (2024).

---

> ### Author Rebuttal · Authors · 2025-07-30
>
> We thank the reviewer for the thorough evaluation of our work and the valuable literature suggestions. We are glad the reviewer found the problem important and the experiments well-designed. Below, we address the raised concerns in detail and **provide additional results** and clarifications. All changes described will be included in the final paper.
>
> ## **Showing Generalization to Novel Subjects and Stimuli from a Different Dataset**
>
> **Experiment on novel stimuli + novel participants:** To evaluate our model’s generalization to both unseen participants and unseen stimuli, we report below the improvement in brain alignment of multi-brain-tuned and single-brain-tuned models on an entirely **new fMRI dataset** that was not used at all in our brain-tuning approach: the Narratives dataset [K]. Results were calculated over **16 participants** using only a **56 mins** stimulus. Furthermore, as a strong baseline, we compare to a Multi-brain-tuned model trained directly on the Narratives participants (i.e., multi-brain-tuning with all of the 16 participants). The table below shows the means and standard error for the percent improvement in brain alignment relative to the pretrained model. The reported P-values are to indicate statistical significance from pretrained (obtained using the paired t-test).
>
>
> |  | **Our Multi-brain-tuned** | **Single-brain-tuned** | **Narratives-multi-brain-tuned** |
> | --- | --- | --- | --- |
> | Improvement Mean ± STE | 10.38 ± 2.69 | 7.53 ± 3.15 | 13.8 ± 3.01 |
> | P-value | 0.0018 | 0.0033 | 0.00011 |
>
> Our brain-tuned models show general improvement on the new fMRI stimuli and participants from the Narratives dataset. Notably, our original Multi-brain-tuned model performs close to the  Narratives-multi-brain-tuned model. This further demonstrates the generalization and efficiency of our method on **novel subjects and stimuli with very limited data**.
>
> ## **Design Choices for Multi-brain-tuning Compared to Alternatives like [C]**
>
> Our work indeed aims to create more **generalizable and efficient** brain-language models. While the method of **unifying different participants** in the same space is one of the most important design choices, it’s not the only one. In the paper, we explored several factors (data presentation/binding, loss function, number of trainable parameters) that significantly impact performance (see **Fig. 6**, Appendix A.3 for the effect of shared vs. separate stimuli presentation; **Fig. 4** for loss and LoRA rank ablations). Here, we provide **new additional analysis** comparing our approach to alternative multi-participant handling strategies:
>
> 1. **Separate projection heads per participant:** We trained a variant of our model where each participant has their own projection layer (similar to the conditional projections in [C]). This drastically increases the number of parameters as participants increase.
> 2. **Shared Response Model (SRM):** We applied the classic SRM [H] to the fMRI data (i.e., instead of FreeSurfer), projecting all participants’ data into a shared space as a pre-processing step, and then trained a model on this aligned data.
> 3. **Non-linear projection head:** We replaced our simple linear projection with a multi-layer perceptron to test if a non-linear mapping of brain features improves alignment.
>
> We measured the brain alignment improvement (relative to the pretrained model) on training and held-out participants for each variant:
>
> |  | **Ours (Multi-brain-tuned)** | **Separate Heads** | **SRM-aligned** | **Non-linear Head** |
> | --- | --- | --- | --- | --- |
> | **Train (mean ± SE)** | **47.22 ± 3.15** | 23.57 ± 3.80 | 21.60 ± 2.55 | 31.29 ± 5.01 |
> | **Held-out (mean ± SE)** | **33.66 ± 3.41** | 23.20 ± 6.76 | 9.80 ± 1.60 | 30.12 ± 5.73 |
>
> **Analysis:** Our Multi-brain-tuned model outperforms these alternatives on train and held-out participants. This highlights that our proposed approach is not only better, but it’s also:
>
> 1. much more **parameter-efficient** than alternatives, which blow up with the number of participants; for instance, performing Multi-brain-tuning with separate heads on the Narratives data above would require 16 projection heads.
> 2. more **controllable** than SRM since we can control which brain areas to tune with, unlike SRM.
> 3. easier to perform across **multiple datasets** and recording devices since FreeSurfer is widely and easily adopted and doesn’t require re-fitting with new data.
>
> Finally, we reiterate that our method aims at finding the best framework to achieve generalizability and efficiency for brain encoding, which we show **does not depend on a single design choice (e.g., using LoRA)** but rather many design choices, for which we test each one against strong alternatives.
>
> ## **More Clarifications on Novelty and Relevant Literature.**
>
> Thank you for pointing out the relevant recent literature on multi-participant generalization in brain decoding. While our work is primarily concerned with brain encoding, we agree that discussing the recent approaches in brain decoding can help contextualize the gap in the literature and highlight our contributions better. We promise to add that in the updated version of the paper, along with the new results mentioned earlier.
>
> However, we want to emphasize that the generalization in brain encoding by adapting the same methods in brain decoding literature can be challenging and might not even make sense at times, as discussed in [I]. Unlike decoding, brain encoding models often fail to generalize effectively to new participants. Popular multi-participant unification approaches, such as SRM and FreeSurfer, substantially degrade encoding performance when applied across subjects (**~40%** drop as shown below; cross-participant encoding models trained on 3 participants and tested on a 4th). Even recent fine-tuning-based approaches like BrainWavLM [J] struggle with cross-subject generalization.
>
>
> | **Method** | **Single participant** | **Cross participant** | **Avg % Change** |
> | --- | --- | --- | --- |
> | SRM (brain alignment) | 0.104 ± 0.013 | 0.059 ± 0.012 | -42.5% |
> | FreeSurfer (brain alignment) | 0.12 ± 0.021 | 0.072 ± 0.009 | -40.2% |
>
> To the extent of our knowledge, our approach is the first to demonstrate substantial improvements in speech and language encoding performance and generalization by fine-tuning with neural data, while simultaneously beating single-participant models on training participants' performance.
>
> We appreciate the reviewer’s positive feedback, insightful comments and for helping us improve the paper. We hope that our revisions and new results have addressed your concerns, and we would be grateful if the reviewer could reflect this in their score.
>
> ## **References:**
>
> [H] Chen et al. (2015). *A Reduced-Dimension fMRI Shared Response Model.*NeurIPS* 2015.
>
> [I] Beliy et al. (2025). The wisdom of a crowd of brains: A universal brain encoder. arXiv preprint arXiv: 2406.12179
>
> [J] Vattikonda et al. (2025). BrainWavLM: Fine-tuning speech representations with brain responses to language. arXiv preprint arXiv:2502.08866
>
> [K] Nastase et al. (2021). The “Narratives” fMRI dataset for evaluating models of naturalistic language comprehension. Sci Data 8, 250.
>
> .

---

> > ### Comment · Reviewer_KUd9 · 2025-08-03
> >
> > Thank you to the authors for their response. I am satisfied by the new analyses, especially the experiment on novel stimuli. Accordingly, the authors have satisfied my main concerns and I have raised my score.

---

> > > ### Author Response · Authors · 2025-08-09
> > >
> > > We sincerely thank the reviewer for the positive response; we will make sure to integrate these results in the final version of the paper.

---

### Official Review · Reviewer_cQaF · 2025-06-30

**Clarity:** 4
**Significance:** 3
**Originality:** 3
**Rating:** 5
**Confidence:** 4

**Summary:**

The paper introduces a simple yet effective way to fine-tune pretrained speech models on fMRI recordings collected from several listeners who heard the same naturalistic stories. By sharing a projection head across subjects and injecting lightweight LoRA adapters, the method (i) boosts brain–model alignment relative to single-subject tuning, (ii) cuts the amount of fMRI needed per participant, and (iii) yields small but consistent improvements on two downstream speech–semantic tasks. Extensive ablations on loss functions and adapter rank support the design choices, and the authors commit to releasing code and models.

**Questions:**

No

**Ethical Concerns:**

["NO or VERY MINOR ethics concerns only"]

**Final Justification:**

As other reviewers have noted, the paper applies the well-established idea of cross-subject generalization from decoding to the encoding context. Still, employing multi-brain tuning for encoding is novel, and the experiments are technically solid. I therefore maintain my Accept rating.

**Quality:**

4

**Strengths And Weaknesses:**

Strengths
- The experimental design follows best practice, including rigorous preprocessing and strong baselines. Evaluation is thorough: the authors plot data-efficiency curves, test generalisation on held-out subjects, and probe design choices through ablations. Reproducibility is well supported by (to-be) open-sourced code. The paper itself is clearly written and well illustrated.

Weaknesses
- I found no substantive weaknesses that would motivate rejection.

---

> ### Author Rebuttal · Authors · 2025-07-30
>
> We sincerely thank the reviewer for the very positive evaluation and thorough understanding of our work. We’re very pleased that the reviewer recognized the clarity, strength, rigor, and effectiveness of our approach - and "found no substantive weaknesses that would motivate rejection." We promise to release the full code publicly alongside the final version of the paper.

---

> ### Comment · Reviewer_cQaF · 2025-08-03
>
> Thank you to the authors for their response. As other reviewers have noted, the paper applies the well-established idea of cross-subject generalization from decoding to the encoding context. Still, employing multi-brain tuning for encoding is novel, and the experiments are technically solid. I therefore maintain my Accept rating.

---

### Official Review · Reviewer_9aJn · 2025-07-03

**Clarity:** 3
**Significance:** 3
**Originality:** 3
**Rating:** 5
**Confidence:** 4

**Summary:**

The authors present a robust model employing a novel method for aligning neural activity with natural language stimuli. They use both brain-tuning methods at both the individual and group levels to sucessfully generalize semantic task representations across participants. Furthermore, this model does not require all participants to be exposed to the same stimuli, and performs well in higher-order regions in the brain.

**Questions:**

The current model demonstrated improvement in frontal and parietal regions (when tuning with Wav2Vec 2.0) but a decrease in auditory regions. Is this an artifact of the task itself, which is cognitively demanding and would need to capture more high-level semantics?

**Ethical Concerns:**

["NO or VERY MINOR ethics concerns only"]

**Limitations:**

Yes

**Quality:**

4

**Strengths And Weaknesses:**

### Pros:

* The model takes into account individual variability across the task and physiology of brain responses by applying brain-tuning to individual participants separately.

* In order to address group-level activity, the model projects fMRI responses of multiple participants into a common space, in addition to individual tuning.

* There is quite a lot of flexibility in stimuli in that participants do not need to been exposed to the same stimuli.

### Cons:

* (Minor comment) The generalizability may be more tuned towards brain responses to complex tasks (narrative comprehension) rather than across low-level information processing, as demonstrated by the improvements in higher-order areas in conjunction to decreased performance in auditory areas. A simulation demonstrating the robustness and generalizability of this model across different semantically complicated tasks could address this minor concern.

---

> ### Author Rebuttal · Authors · 2025-07-30
>
> We thank the reviewer for the positive evaluation and insightful feedback. We’re pleased that the reviewer recognized the strengths of our approach, including its flexibility and robustness across individuals and stimuli. Below, we address the concern regarding generalization to auditory regions in detail and provide **additional supporting analyses**.
>
> ## **Clarification on Performance in Auditory Regions**
>
> The observed slight alignment drop in auditory regions alongside improvements in frontal and parietal regions reflects the specific tuning approach we adopted:
>
> 1. **Reported Layers**: The reported results in our main analysis were averaged over the upper-middle to late layers of Wav2Vec 2.0, known to encode more semantic rather than acoustic information. Thus, these layers are expected to increase in semantic rather than acoustic performance. This is also consistent with single-brain-tuning layer analyses in [1, 2].
> 2. **Included Brain Regions**: Our multi-brain-tuned model predominantly includes higher-level (late language) brain areas (see Supp A.1 for details on the regions). Consequently, improvements naturally concentrate in regions capturing semantic processing, which are much larger than early auditory regions.
>
> ## **Maintaining Acoustic Performance**
>
> To further clarify the impact of Multi-brain-tuning on auditory and acoustic performance, we report below the performance of our Multi-Brain-tuned model on 1) alignment with only the auditory regions, and 2) a low-level acoustic task (namely MFCC prediction detailed in [1]). As a baseline, we compare it to a multi-brain-tuned model that was tuned using *only* the brain auditory regions (rather than both auditory and late language areas).
>
> The tables below show percent improvement relative to the pretrained model for our Multi-brain-tuned vs Auditory Multi-brain-tuned for (i) the brain alignment of early auditory regions and (ii) performance on MFCC prediction; reported results are the mean over early to mid layers of the model (which perform best in acoustic tasks [1]).
>
> - **Table i:**
> |  | **Auditory Multi-Brain-tuned** | **Our Multi-Brain-tuned** |
> | --- | --- | --- |
> | Train participants (mean ± STE) | 4.95 ± 0.13 | 3.465 ± 0.43 |
> | Held-out participants (mean ± STE) | 2.7 ± 0.53 | -1.25 ± 1.4 |
>
> - **Table ii:**
>
> |  | **Auditory Multi-Brain-tuned** | **Our Multi-Brain-tuned** |
> | --- | --- | --- |
> | Pct Improvement | 4.48 | -0.5 |
>
> Our Multi-brain-tuned shows small positive to zero improvement, indicating **no strong negative effects on the acoustic performance** of the model despite being tuned on semantics-dominated data. Auditory Multi-Brain-tuning shows improved performance on both alignment and acoustic performance, consistent with the findings of the single-participant case in [2]. The reason for not having as huge an improvement as in late language areas is both the small size of the auditory cortex and the already high alignment of pretrained models with auditory regions (pretrained speech models are able to predict **~80%** of the estimated explainable variance in auditory regions [3]).
>
> To sum up, substantial performance gains on high-level rather than low-level tasks is expected, but the above analysis demonstrates that the low-level utility of our Multi-brain-tuned model is either unaffected or slightly improved, alleviating the worry of a compromised low-level performance.
>
> We appreciate the reviewer’s constructive comments and evaluation, which we believe will strengthen the manuscript. We will include these additional analyses, alongside detailed ablations exploring alignment improvements across different brain regions, in the final version of the paper.
>
> ## **References**
> [1] Moussa & Toneva (2025). Brain-tuned speech models better reflect speech processing stages in the brain. *Interspeech 2025.*
>
> [2] Vattikonda et al. (2025). BrainWavLM: Fine-tuning speech representations with brain responses to language. arXiv preprint arXiv:2502.08866.
>
> [3] Oota, et al (2024). Speech language models lack important brain-relevant semantics. ACL, 2024.

---

> > ### Comment · Area_Chair_vJyb · 2025-08-08
> > **.**
> >
> > Dear reviewer,
> >
> > The discussion period is ending in 24 hours. Please respond to the author rebuttal and engage in discussion. You must do so before the "Mandatory Acknowledgment". Failure to do so may result in possible penalties.
> >
> > Thanks!
> > Your AC

---

### Author Response · Authors · 2025-08-09
**Summary of the additions inspired by reviewers' suggestions and discussion**

We thank all reviewers for the insightful reviews. We have added several new findings (to be added to the final version of the paper), which we believe increase the clarity and strength of our work. We summarize these additions and their purpose below:

### **1. Generalization to novel stimuli and participants [Reviewer KUd9]**

To evaluate our model’s generalization to both **unseen subjects and unseen stimuli**, we carried out comparisons on an entirely new fMRI dataset -- a subset of the Narratives fMRI dataset. Results were calculated over **16 participants** using only a **56mins** stimulus. Our brain-tuned models show general improvement on the Narratives participants, further supporting the generalization and efficiency of our method on novel subjects and stimuli with very limited data. In fact, our Multi-brain-tuned model performed **on par** with a strong baseline (the brain-tuned model trained on the same 16 Narratives participants).

### **2. Comparisons with other Multi-participant generalization approaches [Reviewers KUd9 and W1Kc]**

We provide additional comparisons about the effect of the method used to handle multi-participant responses (especially from the decoding literature) to show that our method is more effective in generalizing to multi-participant settings and leads to better improvement in brain alignment. Our method outperformed **4 other multi-participant** generalization alternatives while being able to **scale better** with the number of brain-tuning participants and not requiring all participants to **share** **stimuli** like Tang & Huth (2025).

We also show the advantage of our method against popular brain encoding cross-participant approaches like SRM and FreeSurfer, which -**unlike our method**- show around **40%** **drop** in alignment when tested in cross-participant settings.

### **3. Brain-tuning with auditory brain regions only [Reviewer 9aJn]**

Since the brain-tuning data is dominated by semantic regions, there is a concern that acoustic performance might be impacted. To further clarify the impact of Multi-brain-tuning on auditory and acoustic performance, we test its performance compared to a Multi-brain-tuned baseline that was tuned only using the brain's auditory regions. Our Multi-brain-tuned shows small positive to zero improvement, indicating **no negative effects** on the acoustic performance of the model despite being tuned on semantics-dominated data.


Lastly, as promised in the rebuttal, all these results will be integrated in the final version of the manuscript alongside discussing relevant brain decoding literature to contextualize our approach better in the field. We will also publish the public code with the final version of the paper.

We thank all reviewers again for the valuable discussion and suggestions; they helped us clarify our message, provide stronger evidence for our main method, and potentially inspire some future directions!

---

### Note · Authors · 2025-08-15

We would like to thank the reviewers and AC again for their feedback and engagement throughout the discussion period.

To ensure that our additions and clarifications are not missed, we would like to refer to our summary [official comment](https://openreview.net/forum?id=4jgsUhWWaF&noteId=tzD1cXdEyB) posted at the end of the discussion period. In there, we summarize the main additions in response to reviewers' suggestions and concerns. They mainly include:

- More evidence of Multi-brain-tuning generalization and efficiency to **novel stimuli and participants**.
- Highlighting the advantages of our Multi-brain-tuning approach in comparison to alternative methods of unifying Multi-participant responses (e.g., **some popular approaches used in brain decoding**).
- Showing that Multi-brain-tuning has **no negative effects** on the model’s acoustic performance while being tuned on semantically-dominated brain data.

All results will be integrated into the updated version of the paper alongside including a better coverage of decoding literature. We believe all additions will substantially improve the quality of the work and further clarify our contributions.

---

### Decision · Program_Chairs · 2025-09-17

**Decision:**

Accept (poster)

**Comment:**

This paper presents a simple approach for improving Model-Brain alignment in speech models, which involves using the representation from the speech model to predict fMRI responses from multiple participants. This type of "multi-task" alignment is shown to improve performance and data efficiency in two popular speech models, namely Wav2Vec and HuBERT.

This is a well-executed paper that shows that a simple technique helps improvement alignment. It was especially impressive to see that this approach does not hurt downstream performance, and sometimes even helps. While some reviewers initially noted that the use of the same stimuli is a strict restriction, during the rebuttal the authors showed that the method can generalize to new stimuli. One potential weakness identified by a reviewer was with regard to originality. The authors (in my opinion) adequately responded to this point, but I encourage the authors to incorporate this in the camera ready.

Overall, this paper is a clear accept.